# Genetic Variation in *CYP2D6*, *UGT1A4*, *SLC6A2* and *SLCO1B1* Alters the Pharmacokinetics and Safety of Mirabegron

**DOI:** 10.3390/pharmaceutics16081077

**Published:** 2024-08-17

**Authors:** Paula Soria-Chacartegui, Patricia Cendoya-Ramiro, Eva González-Iglesias, Samuel Martín-Vílchez, Andrea Rodríguez-Lopez, Gina Mejía-Abril, Manuel Román, Sergio Luquero-Bueno, Dolores Ochoa, Francisco Abad-Santos

**Affiliations:** 1Clinical Pharmacology Department, Hospital Universitario de La Princesa, Faculty of Medicine, Instituto de Investigación Sanitaria La Princesa (IP), Universidad Autónoma de Madrid (UAM), 28006 Madrid, Spain; 2Centro de Investigación Biomédica en Red de Enfermedades Hepáticas y Digestivas (CIBERehd), Instituto de Salud Carlos III, 28029 Madrid, Spain

**Keywords:** mirabegron, pharmacogenetics, *CYP2D6*, overactive bladder, pharmacokinetics

## Abstract

Mirabegron is a drug used in overactive bladder (OAB) treatment. Genetic variation in pharmacogenes might alter its pharmacokinetics, affecting its efficacy and safety. This research aimed to analyze the impact of genetic variation on mirabegron pharmacokinetics and safety. Volunteers from three bioequivalence trials (n = 79), treated with a single or a multiple dose of mirabegron 50 mg under fed or fasting conditions, were genotyped for 115 variants in pharmacogenes and their phenotypes were inferred. A statistical analysis was performed, searching for associations between genetics, pharmacokinetics and safety. CYP2D6 intermediate metabolizers showed a higher elimination half-life (t_1/2_) (univariate *p*-value (*p*_uv_) = 0.018) and incidence of adverse reactions (ADRs) (*p*_uv_ = 0.008, multivariate *p* (*p*_mv_) = 0.010) than normal plus ultrarapid metabolizers. The *UGT1A4* rs2011425 T/G genotype showed a higher t_1/2_ than the T/T genotype (*p*_uv_ = 0.002, *p*_mv_ = 0.003). A lower dose/weight corrected area under the curve (AUC/DW) and higher clearance (CL/F) were observed in the *SLC6A2* rs12708954 C/C genotype compared to the C/A genotype (*p*_uv_ = 0.015 and 0.016) and ADR incidence was higher when the *SLCO1B1* function was decreased (*p*_uv_ = 0.007, *p*_mv_ = 0.010). The lower elimination and higher ADR incidence when CYP2D6 activity is reduced suggest it might be a useful biomarker in mirabegron treatment. UGT1A4, SLC6A2 and SLCO1B1 might also be involved in mirabegron pharmacokinetics.

## 1. Introduction

Overactive bladder (OAB) syndrome is a condition characterized by unstable bladder contractions, which causes urinary urgency, with or without incontinence, frequent urination and nocturia [1]. Observed in approximately 12% of the population, its incidence and morbidity increase with age, particularly after the age of forty [1,2,3]. It has a highly negative impact on the quality of life, which emphasizes the need for an effective treatment. OAB management is typically based on a combination of behavioral therapies, such as weight loss and pelvic floor exercises, and pharmacological treatment, which includes antimuscarinics (also known as anticholinergics) and β3-agonists [4]. Mirabegron is a first-in-class drug that belongs to the family of β3-adrenoreceptor agonists [5]. It was approved by the European Medicines Agency (EMA) in 2012 for OAB treatment, after it was demonstrated to reduce urinary frequency, incontinence and urgency and to improve health-related quality of life (HR-QOL) [5].

The mechanism of action of mirabegron is the upregulation of the sympathetic nervous activity, which promotes the relaxation of the detrusor muscle of the bladder. Consequently, the storage period before feeling the need to urinate is prolonged [1]. Mirabegron represents a promising pharmacological alternative to anticholinergics, which are associated with several adverse drug reactions (ADRs) that reduce treatment adherence [1]. Furthermore, β3-agonists represent the only effective treatment for OAB when it is caused by the metabolic syndrome or by menopausal hormonal changes [6].

Mirabegron is administered orally following a posology of 25–50 mg every 24 h, with an absolute bioavailability of 29–35% [7]. The maximum plasma concentration (C_max_) is reached 3–4 h (t_max_) after drug intake, showing a plasmatic protein binding of approximately 71%. It is metabolized to inactivate products, potentially by the action of enzymes from the cytochrome P450 (CYP) and the UDP-glucuronosyltransferase (UGT) families, such as CYP2D6, CYP3A4, UGT2B7, UGT1A3 or UGT1A8 [7,8]. Mainly excreted through urine (55%) and feces (34%), its elimination half-life (t_1/2_) is 50 h, while the total body clearance (CL) from plasma is 57 L/h [7].

The most frequently reported ADRs in mirabegron treatment, which are less common than with anticholinergics [1], are headache, tachycardia, urinary tract infection and dizziness [7]. Its occurrence is influenced by several factors, such as sex, age, concomitant medication and genetics [9]. Consequently, pharmacogenetics may provide physicians with effective clues to optimize drug efficacy and safety [9]. Pharmacogenetics is an approach to precision medicine that tailors drug selection and dosing to the patient’s genetic features [10] and it is already implemented in clinical practice for a variety of drugs [11,12,13,14,15,16,17]. However, no pharmacogenetic clinical guidelines are available to direct the treatment with mirabegron or with any antimuscarinic drug [18]. Thus, in order to enhance pharmacogenetic implementation in OAB treatment, it is essential to investigate the effect of genetic variation on these drugs’ behavior and response. Consequently, the main objective of this pharmacogenetic research was to study the impact of genetic variation on mirabegron pharmacokinetics and safety. This research forms part of “La Princesa Multidisciplinary Initiative for the Implementation of Pharmacogenetics” (PriME-PGx) [19].

## 2. Materials and Methods

### 2.1. Study Population

The population included in the research was composed of healthy volunteers from three bioequivalence clinical trials conducted at the Clinical Trials Unit of the Hospital Universitario de La Princesa (UECHUP), Madrid, Spain. Subjects self-reported their biogeographic origin, sex and age. A total of 104 volunteers provided written informed consent to participate in the bioequivalence trials, and 79 of them provided written informed consent to be included in the observational pharmacogenetic study. The protocol and the informed consent form of the three clinical trials and the pharmacogenetic study (code SFC-FG-2020-1, IRB/Code: 4176) were reviewed and approved by the Spanish Drug Agency (AEMPS) and by the Ethics Committee on Clinical Research with Drugs of the Hospital Universitario de La Princesa. Moreover, the studies were conducted in accordance with the guidelines of Good Clinical Practice [20], current Spanish legislation and the Revised Declaration of Helsinki [21].

### 2.2. Clinical Trial Design

The three trials included in this research were phase I, open-label, randomized, replicated and crossover bioequivalence clinical trials of mirabegron 50 mg prolonged-release tablets (Table 1). Bioequivalence trials constitute a specific type of phase I clinical trial in which two formulations of the same drug are compared to ascertain the similarity of their pharmacokinetic profile [22]. These studies are frequently conducted in order to approve the commercialization of a test formulation (T) by demonstrating its bioequivalence to a reference formulation (R). In clinical trials A and B, the administration of a single mirabegron dose under fasting and fed conditions was respectively examined. Fed conditions in clinical trial B were ensured by a high-fat 900 kcal breakfast, in accordance with EMA guidelines for bioequivalence trials [23]. The administration of 12 oral doses in fasting conditions was analyzed in trial C (Table 1).

The inclusion criteria for all the clinical trials included males and females aged from 18 to 55, with no clinically significant organic or psychic conditions and no abnormalities in medical records, in laboratory analysis and in vital signs and electrocardiogram. The exclusion criteria included the use of any prescribed medication in the past 15 days; body mass index (BMI) outside the range 18.5–30 kg/m^2^; a history of drug sensitivity; use of recreational drugs, alcohol or tobacco; blood donation in the past month; pregnant or breastfeeding women; participation in other clinical trials in the past three months; and a history of swallowing problems.

To ensure both formulations were administered twice to each volunteer, clinical trials A and B were structured in four single dose periods, with four different sequences: T-R-T-R, T-R-R-T, R-T-R-T and R-T-T-R. Hospitalization lasted from at least 10 h before to 12 h after the intake of the T or the R formulation (depending on the sequence) in each period. The washout period, i.e., the time between periods, necessary to ensure complete drug elimination, was 14 days. Clinical trial C was also structured in four periods. However, 11 doses (once daily) were administered in the first and the third period, whereas the second and fourth period consisted of the administration of a single dose 24 h after the last administration from the previous period. Thus, there was no washout time and two different sequences were followed: T-T-R-R and R-R-T-T. Subjects were hospitalized from at least 10 h prior to the start of the first and the third period to 12 h following the conclusion of the second and fourth period, respectively.

### 2.3. Pharmacokinetic and Safety Analysis

In clinical trials A and B, at least 16 blood samples were collected from pre-dose to 72 h after drug intake in the four periods. In clinical trial C, between 16 and 21 blood samples were extracted from pre-dose to 24 h after the last dose in each period. In the three clinical trials, blood samples were immediately centrifuged after extraction, and plasma was stored at −20 °C until its shipment to an external laboratory for mirabegron plasmatic quantification. The measurement of mirabegron plasma concentration was performed in accordance with the EMA guidelines by reversed phase ultra-high-performance liquid chromatography coupled to tandem mass spectrometry (UPLC-MS/MS), with a lower limit of quantification (LLOQ) of 100 pg/mL. The study was only blinded for the determination of mirabegron plasmatic concentration.

The estimation of the pharmacokinetic parameters was performed by non-compartmental methods, using the WinNonLin Professional version 8.3 software (Scientific Consulting, Inc., Cary, NC, USA). The maximum plasma concentration (C_max_; C_max-ss_ in steady state for clinical trial C) and the time to reach it (t_max_) were directly obtained from the plasma concentration–time curves. The total area under the curve (AUC_∞_) was calculated as the sum of the AUC from pre-dose to the last observed concentration point (AUC_0–t_), obtained by the trapezoidal method, and the AUC from the t point to infinity (AUC_t–∞_). The AUC_t–∞_ was calculated as the ratio of the last detectable concentration (Ct) divided by the elimination slope (K_e_) of the log-linear portion from the concentration–time curve obtained by linear regression. In trial C, the AUC between dosing intervals (AUC_tau_) was calculated with the trapezoidal method. The t_1/2_ was calculated as ln2/K_e_ and the total CL adjusted for bioavailability (CL/F) was calculated as the dose divided by AUC_∞_ and weight. Both formulations were deemed to be bioequivalent in the three trials; thus, the arithmetic mean of the parameters obtained in the four periods was calculated for each subject.

The occurrence of adverse effects (AEs) was documented through direct questioning of the subjects and spontaneous reporting. The algorithm of the Spanish Pharmacovigilance System was applied to assess AE causality [24]. The AEs with a definite, probable or possible causal relationship with mirabegron were considered ADRs.

### 2.4. Genotyping and Phenotyping

DNA was extracted from the blood samples collected during the clinical trials using a Maxwell^®^ RSC Automated DNA extractor (Promega Biotech Iberica S.L, Madrid, Spain). The customized array PriME-PGx Very Important Pharmacogene Open Array (VIPOA), based on TaqMan^®^ probes, was used for genotyping in a QuantStudio 12 k Flex real-time PCR System (Applied Biosystems, ThermoFisher, Waltham, MA, USA) with an Open Array thermal block. A total of 115 genetic variants from genes codifying for metabolizing enzymes and transporters were analyzed (Appendix A). Nucleotide variant annotation was performed according to the National Center for Biotechnology Information (NCBI) Reference Sequence (RefSeq) database [25]. The sample size available for each genetic variant differs due to the occurrence of low-frequent genotyping failures.

The presence of duplications, deletions or hybrid structures in the gene *CYP2D6* was studied. A copy number variation assay for both ends of the gene (5′ untranslated region and exon 9) was performed in the QuantStudio 12 k Flex real-time PCR System (Applied Biosystems, ThermoFisher, Waltham, MA, USA) with a 96-well thermal block [26].

The Clinical Pharmacogenetics Implementation Consortium (CPIC) and the Pharmacogene Variation Consortium (PharmVar) database were used for allele definition [27,28]. The pharmacogenetic phenotype for *CYP2B6* [13], *CYP2C19* [11], *CYP2C9* [12], *CYP2D6* [14], *CYP3A4* [29], *CYP3A5* [15], *NUDT15* [16], *SLCO1B1* [12], *TPMT* [16] and *UGT1A1* [17] was inferred from diplotypes based on their corresponding CPIC and Dutch Pharmacogenetic Working Group (DPWG) guidelines. For *CYP2C8* and *NAT2*, phenotype was inferred as described in previous works [30,31]. The remaining genetic variants were analyzed individually.

### 2.5. Statistical Analysis

SPSS Statistics software version 29.0 (SPSS Inc., Chicago, IL, USA) was used for the statistical analysis. To correct the effect of dose and weight on exposure, the AUC_∞_ (trials A and B), the AUC_tau_ (trial C) and the C_max_ were divided by the dose/weight ratio (DW). The normality of the dependent variables was analyzed by the Shapiro–Wilk test. Those variables not following a normal distribution were logarithmic-transformed, and normality was re-analyzed. The significance level was established at *p*-value (*p*) < 0.05. The demographic characteristics (age, height, weight and BMI) were analyzed according to sex, biogeographic origin and clinical trial design.

First, a univariate pharmacokinetic analysis was performed. Pharmacokinetic parameters were analyzed according to sex, biogeographic origin, clinical trial design, genotypes and phenotypes. For the analysis of normally distributed dependent variables according to independent variables with two or with three or more categories, *t*-tests and ANOVA tests followed by Bonferroni post hoc tests were respectively performed. For the analysis of not normally distributed dependent variables according to independent variables with two or with three or more categories, Mann–Whitney tests and Kruskal–Wallis test were conducted, respectively. Afterwards, a multivariate pharmacokinetic analysis was performed by means of linear regression. Those independent variables with significant differences in the univariate analysis (univariate *p*-value (*p*_uv_) < 0.05) were included. Significant associations are shown as the multivariate *p*-value (*p*_mv_), the standardized β coefficient (β) and the R^2^.

For safety evaluation, differences in the AUC and C_max_ were analyzed in relation to ADR incidence using the *t*-test and the Mann–Whitney test, respectively. In the univariate safety analysis, ADR incidence was analyzed according to sex, biogeographic origin, clinical trial, genotypes and phenotypes by a Pearson’s Chi^2^ test or a Fisher’s exact test when the expected frequency in at least 20% of cells was lower than 5. Subsequently, a multivariate safety analysis was conducted by means of logistic regression. The independent variables significantly associated with ADR incidence in the univariate analysis (*p*_uv_ < 0.05) were included. Significance is shown as *p*_mv_ and odds ratio (OR).

## 3. Results

### 3.1. Demographic Characteristics

The height and weight of females and Latin Americans were lower than in males (*p* < 0.001 and *p* < 0.001, respectively) and Europeans (*p* = 0.005 and *p* = 0.01, respectively) (Table 2). No differences were identified in the demographic parameters according to the clinical trial (Table 2).

### 3.2. Pharmacokinetics

Females showed a higher AUC and C_max_ (455.74 ± 152.50 ng·h/mL and 47.25 [20.95–57.58] ng/mL) compared to males (354.88 ± 120.42 ng·h/mL and 33.11 [18.63–44.88] ng/mL, *p* = 0.002 and *p* = 0.009, respectively); however, the differences disappeared after DW correction (Table 3).

The AUC/DW and the C_max_/DW were lower in clinical trial B compared to the other clinical trials (AUC/DW: *p*_uv_ < 0.001 compared to trials A and C, *p*_mv_ < 0.001, β = 0.668, R^2^ = 0.438; and C_max_/DW: *p*_uv_ < 0.001 compared to trials A and C, *p*_mv_ < 0.001, β = 0.739, R^2^ = 0.539). Conversely, the t_max_ and the CL/F were higher (t_max_: *p*_uv_ = 0.014 compared to trial A, *p*_uv_ < 0.001 compared to trial C, *p*_mv_ < 0.001, β = −0.426, R^2^ = 0.169; and CL/F: *p*_uv_ < 0.001 compared to trials A and C, *p*_mv_ < 0.001, β = −0.691, R^2^ = 0.469). Furthermore, the t_1/2_ in clinical trial C was lower than in clinical trials A and B (*p*_uv_ < 0.001 compared to trials A and B, *p*_mv_ < 0.001, β = 0.535, R^2^ = 0.381). No differences in the pharmacokinetic parameters according to biogeographic origin were observed (Table 3).

### 3.3. Pharmacogenetics

No CYP2D6 poor metabolizers (PMs) were available in this research. A higher t_1/2_ was observed in CYP2D6 intermediate metabolizers (IMs) compared to ultrarapid (UMs) and normal metabolizers (NMs) (*p*_uv_ = 0.028). Significance was transformed into a tendency after Bonferroni post hoc tests (NMs vs. IMs, *p*_uv_ = 0.074, UMs vs. IMs *p*_uv_ = 0.124 and NMs vs. UMs *p*_uv_ = 0.624). However, the t_1/2_ remained significantly higher in IMs compared to UMs and NMs merged in a unique group (UMs + NMs) (*p*_uv_ = 0.018). Differences in the t_1/2_ were accompanied by a not significantly lower AUC/DW and C_max_/DW in CYP2D6 UMs compared to NMs and IMs. The *UGT1A3* rs2008584 A/A genotype was associated with lower t_max_ than the *UGT1A3* rs2008584 A/G genotype (*p*_uv_ = 0.014). Also, the t_1/2_ was higher in the *UGT1A4* rs2011425 T/G genotype compared to the T/T genotype (*p*_uv_ = 0.002, *p*_mv_ = 0.003, β = 0.268, R^2^ = 0.381) (Table 4).

The C_max_/DW was significantly lower in NAT2 slow acetilators (SAs) compared to intermediate (IAs) and rapid acetilators (RAs) (*p*_uv_ = 0.046). Even though differences disappeared in pair comparison, the observed lower C_max_/DW in NAT2 SAs was maintained when IAs and RAs were merged in a unique group (*p*_uv_ = 0.016), accompanied by a tendency towards a lower AUC/DW (*p*_uv_ = 0.089) and higher t_max_ and CL/F in SAs (*p*_uv_ = 0.083 and *p*_uv_ = 0.076, respectively).

A lower AUC/DW and a higher CL/F were observed in the *SLC6A2* rs12708954 C/C genotype compared to the C/A genotype (*p*_uv_ = 0.015 and *p*_uv_ = 0.016, respectively). Lastly, a lower C_max_/DW was observed in the *SLC19A1* rs1051266 A/A genotype compared to the A/G genotype (*p*_uv_ = 0.016) (Table 4). For *CYP3A4*, the AUC/DW and C_max_/DW were not significantly higher in IMs compared to NMs (Appendix A). No further differences were observed in the pharmacokinetic parameters according to the remaining genetic variants (Appendix A).

### 3.4. Safety

ADRs were present in ten volunteers (12.66%) in total (two in clinical trials A and C, and six in clinical trial B, Table 5), five of whom suffered one ADR, three of whom suffered two ADRs and two of whom suffered three ADRs. Headache was the most common ADR (9 times), followed by nausea (3 times), vomiting (2 times), loose stool, palpitations and insomnia (1 time each). ADR incidence was similar between males and females (7.90% vs. 17.07%, *p* = 0.314), between Europeans and Latin Americans (5.60% vs. 14.80%, *p* = 0.414) and among clinical trials A, B and C (9.10% vs. 20.00% vs. 7.40%, respectively, *p* = 0.331). The AUC and C_max_ in volunteers suffering ADRs and volunteers not experiencing them were not significantly different when the three trials were analyzed collectively and individually, with the exception of a tendency towards a higher AUC in the volunteers experiencing ADRs compared to those with no ADRs in clinical trial B (*p* = 0.070) (Table 5).

CYP2D6 IMs showed a higher ADR incidence (37.5%) than NMs (5.5%) and UMs (0%) (*p*_uv_ = 0.008, *p*_mv_ = 0.010, OR = 9.83). Also, ADRs were more frequent in volunteers with decreased function (DF) for SLCO1B1 (30.4%) compared to those with normal function (NF) (5.7%) (*p*_uv_ = 0.007, *p*_mv_ = 0.010, OR = 11.24). No further differences in ADR incidence according to genotypes or phenotypes were observed.

## 4. Discussion

The successful treatment of the OAB condition is of great relevance due to its negative impact on quality of life and healthcare systems and its indirect costs due to the loss of work capacity [32,33]. Using mirabegron for OAB treatment could potentially decrease the global treatment costs in comparison to antimuscarinic drugs [34]. However, the effect of genetic variation on mirabegron response is still uncertain. Thus, this research aimed to identify genetic factors explaining variability in mirabegron pharmacokinetics and safety.

The pharmacokinetic parameters observed in this research were consistent with those previously reported. The AUC, C_max_ and t_max_ described in previous works with mirabegron ranged between 315–350 ng·h/mL, 16.1–39.2 ng/mL and 3–5 h, respectively, which are comparable to the results observed in this research [7,35,36,37]. Conversely, the t_1/2_ was lower than the 50 h indicated in the drug label [7]. However, a mirabegron t_1/2_ of 23–25 h and 40 h were also reported in other investigations [38,39]. It is likely that these differences are caused by the fact that the last sample in this research was extracted 72 h after drug intake, whereas a precise estimation of mirabegron t_1/2_ requires the collection of samples at least until 150 h after drug intake (i.e., three t_1/2_).

The lower exposure observed in females compared to males disappeared upon DW correction. Therefore, it is likely that these differences are caused by the lower weight of women compared to men, as shown in the demographic analysis, rather than by other sex differences. Actually, this higher exposure in women was previously documented in other studies with mirabegron [35,36,37]. If these differences are found to be clinically relevant in further research, the potential of adjusting dose to weight, rather than prescribing a unified dose, should be considered. Furthermore, a lower mirabegron exposure and a higher t_max_ were observed in fed volunteers (trial B) compared to those in fasting conditions (trials A and C). Consistently, it has been proposed that high-fat and low-fat meals reduce mirabegron AUC and C_max_ and increase its t_max_ [7,40]. Since different transporters are involved in mirabegron influx into the lumen, such as the organic anion transporting polypeptide 2A1 (OATP2A1), the lower exposure and absorption observed may be attributable to the inhibition of these transporters by food [41,42]. The delayed mirabegron absorption in the presence of food might be explained by a delayed gastric emptying [36]. Despite these pharmacokinetic differences, safety and efficacy were achieved under both fed and fasting conditions in previous research [7]. Consequently, these differences are expected to be clinically irrelevant, and no dose adjustment ought to be required.

Mirabegron is extensively metabolized in the liver by CYP3A4 and CYP2D6 [43]. In this research, a higher t_1/2_ was observed in CYP2D6 IMs compared to NMs and UMs, and a progressive reduction in the AUC/DW, C_max_/DW and t_1/2_ and a progressive increase in CL/F when CYP2D6 activity augmented was observed. The lack of significance in exposure and CL might be caused by the reduced sample size available: only two UMs were included, with zero PMs. Indeed, to the best of our knowledge, differences in mirabegron exposure were previously only reported in CYP2D6 PMs, which showed higher drug exposure than NMs [18]. In accordance, individuals with a higher t_1/2_ (i.e., CYP2D6 IMs) presented a higher incidence of ADRs in this research. Thus, the augmented elimination rate in volunteers with higher CYP2D6 activity, along with their lower ADR incidence, suggests that CYP2D6 is involved in mirabegron metabolism and it might be a useful biomarker in the treatment with this drug. Nevertheless, a low contribution of CYP2D6 in mirabegron pharmacokinetics was proposed in previous research [44], which may also explain the lack of significance in CL and exposure. Consequently, further studies with increased sample size are required to study the impact of the CYP2D6 phenotype on the pharmacokinetics and safety of mirabegron.

On the other hand, no significant differences in mirabegron pharmacokinetics were observed based on the CYP3A4 phenotype, although it is proposed as its main metabolizing enzyme [45]. The exposure was higher in IMs than in NMs (see Appendix A), albeit not significantly, probably due to the reduced sample size; only two volunteers were CYP3A4 IMs and no PMs were found. Furthermore, the high inductility and inhibition of CYP3A4, and the different *CYP3A4* expression levels between the sexes, may neutralize the impact of *CYP3A4* genetic variation on its enzymatic activity [46,47]. Nevertheless, it should also be considered that other enzymes, such as CYP2D6 and UGTs, might play a more relevant role than CYP3A4 in mirabegron metabolism. Consequently, further research is needed to ascertain the relevance of *CYP3A4* genetic variation in mirabegron pharmacokinetics.

Genetic variation in *UGT1A3* and *UGT1A4* was found to alter mirabegron pharmacokinetics in this study. A higher t_1/2_ was observed when the *UGT1A4* rs2011425 T/G genotype was present, suggesting decreased elimination compared to the T/T genotype. This missense genetic variant was previously associated with both increased and decreased UGT1A4 glucuronidation activity in research with other drugs [48,49]. However, to the best of our knowledge, no previous associations between this genetic variant and differences in mirabegron pharmacokinetics or response are known. Nevertheless, other enzymes from the *UGT1A* gene cluster do participate in mirabegron metabolism, and it is known that enzymes from this family present substrate overlap [50]. Therefore, it could be proposed that the G allele in *UGT1A4* rs2011425 causes a decreased mirabegron metabolization compared to the T allele, increasing the time needed for its elimination. However, a different study determined that the role of UGT1A4 in mirabegron metabolism is negligible [51]. Secondly, *UGT1A3* rs2008584 was only associated with differences in t_max_, which is a pharmacokinetic parameter more dependent on transporter activity, rather than that of hepatic metabolizing enzymes. In light of these considerations and the results obtained, it could be proposed that UGT1A4 rather than UGT1A3 might be involved in mirabegron metabolism, since genetic variation in *UGT1A4* altered mirabegron t_1/2_. Nevertheless, further research on this association and on the impact of genetic variation in UGT1A4 activity should be performed. Lastly, *UGT2B7* and *UGT1A8* genetic variation was not found to impact mirabegron pharmacokinetics (see Appendix A). However, genetic variation in these genes and in their gene family is not well characterized (e.g., the impact of known genetic variants on enzyme activity is not clear and no haplotypes and phenotypes have been identified and defined). Therefore, a better structural and functional characterization of the genetic variation in these *UGT* genes is needed prior to evaluating their relevance as pharmacogenetic biomarkers.

The gene *NAT2* codifies for the enzyme N-acetyltransferase 2, which is involved in the metabolism of diverse drugs [31,52,53]. A lower mirabegron C_max_/DW in individuals with lower NAT2 activity was observed. These results are contrary to previous results in the literature with other drugs, in which lower NAT2 activity was associated with reduced drug elimination and therefore higher exposure [54]. The observed differences might be caused by the different distribution of RAs, IAs and SAs in the three clinical trials. Actually, individuals with higher NAT2 activity were only present in clinical trials A and C, in which drug exposure was increased, compared to clinical trial B (Trial B: 0% RAs, 34.48% IAs and 65.52% SAs versus trials A + C: 14.29% RAs, 48.87% IAs and 36.73% SAs, *p* = 0.007). The lack of significant differences in the multivariate analysis supports this hypothesis. Nevertheless, the NAT2 role in mirabegron pharmacokinetics should be investigated in further research once the impact of each allele is clearly informed in the PharmVar database [55].

The solute carriers (SLC) transporters SLC6A2, SLC19A1 and SLCO1B1 were associated with differences in mirabegron pharmacokinetics and safety, respectively. Volunteers with the *SLC6A2* rs12708954 C/C genotype presented lower exposure compared to those with the C/A genotype. *SLC6A2* codifies for a noradrenaline transporter (NET) and NET transporters are involved in mirabegron uptake into the sympathetic nerve terminals [56]. Indeed, the *SLC6A2* rs12708954 C/C genotype was previously associated with higher response and ADR incidence in other drugs that are also NET substrates, such as atomoxetine [57]. Despite the *SLC6A2* expression in different regions of the gastrointestinal tract [58], it is unknown whether it has a role in drug pharmacokinetics. For these reasons and considering the reduced sample size available, further research is required to determine the relevance of this transporter to mirabegron pharmacokinetics. On the other side, SLC19A1 is a folate transporter with no previous associations with mirabegron pharmacokinetics [7,41,59]. Considering that the differences were observed in the heterozygous genotype, and their disappearance in the multivariate analysis, this result ought to be regarded as spurious.

Lastly, *SLCO1B1* codifies for the OATP1B1, which is a liver specific transporter that contributes to the hepatic uptake of many drugs [60]. A higher ADR incidence in volunteers with lower SLCO1B1 activity was observed. To the best of our knowledge, no studies have proposed that OATP1B1 is involved in mirabegron transit, but previous research demonstrated that OATP1A2, a transporter of the same family, was capable of transporting mirabegron [41]. In research with other drugs, such as statins, a decreased SLCO1B1 activity was related to higher exposure, potentially reaching toxic concentrations that led to the appearance of ADRs [12]. However, differences in ADRs are expected to be accompanied by pharmacokinetic differences, which did not occur. Notwithstanding, when data from each clinical trial were analyzed individually, a trend towards a higher AUC/DW was observed in SLCO1B1 DFs (439.35 ± 125.50 kg·ng·h/mL·mg, n = 13) compared to NFs (364.01 ± 66.01 kg·ng·h/mL·mg, n = 15, *p*_uv_ = 0.080) in clinical trial B (fed conditions), but not in clinical trials A and C (fasting conditions). Thus, it could be proposed that mirabegron hepatic uptake is modified in the presence of food, conferring a more relevant role to SLCO1B1 under these conditions. Certain foods and food additives are known to inhibit different OATP isoforms [42], such as OATP2B1 [61], but, to our knowledge, no correlation between food consumption and SLCO1B1 inhibition has been proposed. Further research on the role of SLCO1B1 in mirabegron transport and its interaction with food is required.

This study presents some limitations. First, the sample size was arbitrary and limited, as it was determined by the amount of data available from the clinical trials. Increased sample size in further research would be advisable to ensure the not inconsiderable presence of low-prevalence genotypes and phenotypes (e.g., CYP2D6 UMs and PMS or CYP3A4 PMs), which would improve the statistical power of the study. Secondly, the genetic variants analyzed were selected based on previous knowledge and on their role in drug pharmacokinetics. Therefore, there is risk of missing unknown variants present in the genome that could alter mirabegron pharmacokinetics. Moreover, the subjects were healthy volunteers, and the majority of them received a single mirabegron dose, which reduced population variability, prevented efficacy assessment and hampered the safety analysis due to the low incidence of ADRs. Nevertheless, this research also offered advantages. It was performed in a controlled environment that allowed avoiding confounding factors, e.g., smoking, comorbidities or concomitant treatment, which are present in investigation with patients. Additionally, correct compliance with medication intake, homogeneous feeding conditions and controlled and precise sampling were ensured.

## 5. Conclusions

Having a reduced CYP2D6 activity was associated with lower elimination rate and higher incidence of ADRs, a finding which paves the way towards the use of this biomarker in mirabegron treatment. The metabolizing enzyme UGT1A4 and the transporters SLC6A2 and SLCO1B1 might be involved in mirabegron metabolism and absorption and distribution, respectively. The impact of *CYP3A4* and *UGT2B7* genetic variation in mirabegron pharmacokinetics requires further research.

## Figures and Tables

**Table 1 pharmaceutics-16-01077-t001:** Characteristics of the clinical trials included in the pharmacogenetic study.

Clinical Trial	EUDRA-CT	Conditions	Dose	Sample Size * (BE Study)	Sample Size * (PG Study)
A	2020-004710-35	Fasting	Single, 50 mg	36 (19/17)	22 (12/10)
B	2021-000283-29	Fed	Single, 50 mg	36 (18/18)	30 (15/15)
C	2021-000285-15	Fasting	Multiple, 50 mg every 24 h	36 (19/17)	27 (11/16)

* Sample size is shown as total number (number of males/number of females). BE: bioequivalence, PG: pharmacogenetic. In each clinical trial, both formulations (T and R) were administered to every volunteer under the conditions and at the dose specified.

**Table 2 pharmaceutics-16-01077-t002:** Demographic characteristics according to sex, biogeographic origin and clinical trial.

	n	Age Years	Height m	Weight kg	BMI kg/m^2^
*Sex*					
Male	38	28.50 (24.00–34.25)	1.76 (0.06)	75.96 (11.64)	24.85 (22.30–27.20)
Female	41	30.00 (24.00–33.50)	1.62 (0.06) *	63.92 (10.21) *	24.33 (21.88–27.37)
*Origin*					
European	18	26.00 (23.75–31.75)	1.73 (0.09)	76.81 (14.61)	26.24 (22.40–27.82)
Latin American	61	31.00 (25.00–34.50)	1.67 (0.09) *	67.61 (10.99) *	24.33 (21.84–26.55)
*Trial*					
A	22	25.00 (23.00–31.50)	1.70 (0.09)	68.30 (11.71)	24.49 (21.04–26.38)
B	30	28.50 (24.00–43.00)	1.68 (0.09)	68.63 (12.32)	23.66 (21.01–27.53)
C	27	32.00 (26.00–33.00)	1.68 (0.10)	72.06 (13.21)	25.64 (23.15–27.58)
*Total*	79	30.00 (24.00–34.00)	1.68 (0.09)	69.71 (12.43)	24.58 (22.12–27.25)

Results are shown as mean (standard deviation) for normally distributed variables and as median (interquartile range) for not normally distributed variables. BMI: body mass index; *: *p* < 0.01 compared to males or Europeans.

**Table 3 pharmaceutics-16-01077-t003:** Pharmacokinetic data according to sex, biogeographic origin and clinical trial.

	n	AUC/DW kg·ng·h/mL·mg	C_max_/DW kg·ng/mL·mg	t_max_ h	t_1/2_ h	CL/F L/kg·h
*Sex*						
Female	41	577.02 (189.54)	60.84 (29.19–74.58)	4.50 (3.94–5.31)	26.50 (8.21)	1.81 (1.44–2.41)
Male	38	529.27 (177.04)	48.31 (30.22–67.24)	4.44 (3.76–5.13)	26.34 (5.22)	2.03 (1.64–2.61)
*Origin*						
European	18	582.84 (202.00)	54.80 (40.65–72.87)	4.69 (4.19–4.91)	25.27 (5.16)	1.75 (1.55–2.33)
Latin American	61	545.56 (179.29)	48.74 (27.06–71.30)	4.41 (3.75–5.31)	26.75 (7.33)	1.92 (1.52–2.57)
*Trial*						
A	22	633.34 (149.93)	58.58 (50.19–67.74)	4.31 (3.97–5.00)	27.63 (3.89)	1.77 (1.40–1.94)
B	30	397.84 (106.91) *	24.85 (19.40–33.98) *	5.13 (4.41–6.25) *	30.62 (7.04)	2.72 (2.21–3.35) *
C	27	663.03 (157.77)	73.88 (63.14–93.71)	4.25 (3.63–4.50)	20.77 (4.51) ^$^	1.64 (1.26–1.81)
*Total*	79	554.06 (184.03)	50.72 (29.50–72.54)	4.50 (3.88–5.13)	26.42 (6.90)	1.91 (1.52–2.51)

Results are shown as mean (standard deviation) for normally distributed variables and as median (interquartile range) for not normally distributed variables. AUC/DW: dose/weight corrected area under the concentration-time curve; C_max_/DW: dose/weight corrected maximum drug plasma concentration; t_max_: time to reach C_max_; t_1/2_: elimination half-life; CL/F: drug clearance adjusted by bioavailability. *: *p*_uv_ < 0.05 compared to clinical trials A and C; ^$^: *p*_uv_ < 0.05 compared to clinical trials A and B; underlined: *p*_mv_ < 0.05.

**Table 4 pharmaceutics-16-01077-t004:** Associations between genotypes or phenotypes and pharmacokinetic parameters.

Gene	Genotype/Phenotype		n	AUC/DW	C_max_/DW	t_max_	t_1/2_	CL/F
				kg·ng·h/mL·mg	kg·ng/mL·mg	h	h	L/kg·h
*CYP2D6*	Phenotype	UM	2	451.77 (231.42)	45.37 [19.63–71.11]	4.94 [4.75–5.13]	20.42 (3.90)	2.60 [1.64–3.55]
		NM	55	547.97 (199.06)	48.74 (28.59–72.57)	4.38 (3.75–5.13)	25.80 (6.01)	1.94 (1.45–2.71)
		IM	18	585.61 (188.13)	50.24 (36.42–67.93)	4.88 (3.97–5.78)	29.97 (8.68) *	1.73 (1.50–2.70)
*UGT1A3*	rs2008584	A/A	16	548.38 (162.14)	58.17 (31.72–83.83)	4.06 (3.38–4.48) *	24.35 (6.62)	1.84 (1.67–2.49)
		A/G	34	564.37 (212.65)	47.89 (25.47–72.54)	4.69 (3.94–5.66)	25.70 (5.44)	1.84 (1.33–2.74)
		G/G	26	537.28 (170.40)	48.67 (28.80–67.93)	4.69 (3.88–5.16)	29.03 (8.39)	1.92 (1.64–2.46)
*UGT1A4*	rs2011425	T/T	64	541.38 (187.75)	48.49 (28.39–73.09)	4.50 (3.75–5.25)	25.28 (6.67)	1.93 (1.55–2.69)
		T/G	14	587.99 (147.20)	57.32 (30.52–66.27)	4.31 (3.97–4.88)	31.14 (5.96) *	1.75 (1.52–1.98)
*NAT2*	Phenotype	RA	7	602.63 (159.64)	60.84 (48.74–65.11)	4.00 (3.26–4.25)	27.80 (7.77)	1.75 (1.68–1.93)
		IA	34	589.33 (196.52)	61.37 (36.10–76.70)	4.38 (3.75–5.16)	24.61 (4.98)	1.81 (1.33–2.38)
		SA	37	516.32 (173.40)	40.91 (24.85–68.97) *	4.75 (4.06–5.56)	27.74 (8.04)	2.19 (1.66–2.67)
*SLC6A2*	rs12708954	C/C	56	522.63 (182.61) *	48.31 (24.23–70.35)	4.45 (3.75–5.09)	26.56 (7.34)	1.93 (1.65–2.81) *
		C/A	21	648.45 (162.98)	65.11 (41.05–78.25)	4.63 (3.94–5.56)	26.29 (6.01)	1.67 (1.31–1.99)
		A/A	2	442.73 (95.66)	47.11 [36.51–57.72]	5.06 [4.88–5.25]	23.94 (2.07)	2.35 [1.98–2.71]
*SLC19A1*	rs1051266	A/A	16	498.20 (190.51)	37.45 (20.53–53.46) *	4.50 (4.03–6.25)	27.92 (5.87)	2.31 (1.50–3.26)
		A/G	34	598.00 (181.87)	60.74 (41.65–74.22)	4.33 (3.75–5.03)	25.72 (5.63)	1.81 (1.43–2.21)
		G/G	27	537.43 (182.46)	48.18 (23.61–71.11)	4.50 (3.76–5.13)	26.12 (8.87)	1.92 (1.61–2.74)

Results are shown as mean (standard deviation) for normally distributed variables and median (interquartile range) for not normally distributed variables. For categories with two volunteers, the two values in square brackets are shown instead of the interquartile range. AUC/DW: dose/weight corrected area under the concentration-time curve; C_max_/DW: dose/weight corrected maximum drug plasma concentration; t_max_: time to reach C_max_; t_1/2_: elimination half-life; UM: ultrarapid metabolizers; NM: normal metabolizers; IM: intermediate metabolizers; RA: rapid acetilators, IA: intermediate acetilators, SA: slow acetilators. *: *p*_uv_ < 0.05 compared to CYP2D6 UMs + NMs, *UGT1A3* rs2008584 A/G genotype, *UGT1A4* rs2011425 T/T genotype, NAT2 RAs + IAs, *SLC6A2* rs12708954 C/A genotype and *SLC19A1* rs1051266 A/G genotype; underlined: *p*_mv_ < 0.05.

**Table 5 pharmaceutics-16-01077-t005:** AUC and C_max_ in volunteers with or without ADRs.

Clinical Trial	ADRs	n	AUC ng·h/mL	C_max_ ng/mL
A	no	20	470.43 (108.04)	45.42 (34.53–51.28)
	yes	2	449.80 (19.61)	44.90 [43.71- 46.09]
B	no	24	282.60 (85.10)	16.81 (12.28–24.08)
	yes	6	359.96 (92.85)	20.28 (12.15–24.60)
C	no	25	476.37 (161.56)	52.75 (43.61–63.79)
	yes	2	505.72 (22.20)	54.95 [51.33–58.57]
Total	no	69	407.25 (152.77)	26.21 (17.99–47.39)
	yes	10	407.08 (94.52)	40.52 (20.99–51.82)

Results are shown as mean (standard deviation) for normally distributed variables and median (interquartile range) for not normally distributed variables. For categories with two volunteers, the two values in square brackets are shown instead of the interquartile range. ADR: adverse reactions; AUC: area under the concentration-time curve, C_max_: maximum drug plasma concentration.

## Data Availability

Data belong to the clinical trials’ sponsors and may be accessible upon reasonable request to the corresponding authors. The data are not publicly available due to the instruction of clinical trial’s sponsors.

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
