# Peer review of "Genetic Variation in CYP2D6, UGT1A4, SLC6A2 and SLCO1B1 Alters the Pharmacokinetics and Safety of Mirabegron"

_pharmaceutics, 2024, doi:10.3390/pharmaceutics16081077_

Round 1

Reviewer 1 Report

Comments and Suggestions for Authors

The authors analyzed data from three bioequivalence clinical trials to examine genetic variations in CYP2D6, UGT1A4, SLC6A2, and SLCO1B1, and their associations with alterations in the pharmacokinetics and safety of mirabegron. The analysis was conducted correctly, and the limitations and potential conflicts of interest are appropriately noted. I recommend this paper for publication, with some very minor adjustments to the text.

Small issues:

1.      The absence of line numbers makes the review process more difficult.

2.       Abstract: The phrase “CYP2D6 intermediate metabolizers showed higher elimination half-life (t1/2) and incidence of adverse reactions (ADRs) than normal plus ultrarapid metabolizers (uni variate p-value (puv)=0.018; puv=0.008, multivariate p (pmv)=0.010, respectively)” is not immediately clear in the part of significances. Can you simplify and clarify what p-values refer to?

3.      Page 3, lines 1-2 from the top: Test formulation (T) and reference formulation ( R) are mentioned. Indicate, which exactly trial/condition/dose is T and which is R in the Table 1.

4.      Page 4, line 1 in the top. Define formally Tmax and C max in this chapter.

5.      In the text, immediately before the Table 2 significances p<0.001 are indicated, then p = 0.005 and p=0.01, but in the Table 2 “*” indicates p<0.05. May be indicating lower p in the Table legend will be in favor of the authors (in case the information in the text is correct).

6.      Table 2, 3rd line: Height, the value in parentless 0.87 is typo or error, I think it must be 0.087

7.      Table 3, 2rd line: CL/F, the value 20.03 is typo or error, I think it must be 2.003

8.      Thus carefully double check all data in the Tables for the typos.

9.      Table 4 title: actual “Statistically significant associations between genotypes or phenotypes and …” may be just “Associations between genotypes or phenotypes and …”, because not all associations are significant in this table.

10. Page 4, line 6 from the bottom. “ADRs were present in ten volunteers…”. It is not clear immediately, these 10 cases referred for A, B or C trial? Indicate it in the text (even, then after it could be concluded from the Table 10).

11. Discussion, page 10, line 18. Literature indicate that UGT1A4 have a negligible role in mitabegron metabolism [5], but in this paper authors show UGT1A4 to have a role in mitabegron pharmacokinetics. Please, comment/speculate more deeply this.

12. For the references [5], [21], please provide the date non only for the availability, but also the date of original reports update (if these reports from 2000, 2010, 2024?)

13. Reference [21] The phrase “World medical association” looks doubled?

Author Response

The authors analyzed data from three bioequivalence clinical trials to examine genetic variations in CYP2D6, UGT1A4, SLC6A2, and SLCO1B1, and their associations with alterations in the pharmacokinetics and safety of mirabegron. The analysis was conducted correctly, and the limitations and potential conflicts of interest are appropriately noted. I recommend this paper for publication, with some very minor adjustments to the text.

Thank you very much for your revision and your commentaries. We appreciate the good valoration of the manuscript and the proposed modifications to improve its quality.

 Small issues:

  1. The absence of line numbers makes the review process more difficult.

Thank you for your observation and sorry for the inconvenience. We have included line numbers in the revised version of the manuscript.

  1. Abstract: The phrase “CYP2D6 intermediate metabolizers showed higher elimination half-life (t1/2) and incidence of adverse reactions (ADRs) than normal plus ultrarapid metabolizers (uni variate p-value (puv)=0.018; puv=0.008, multivariate p (pmv)=0.010, respectively)” is not immediately clear in the part of significances. Can you simplify and clarify what p-values refer to?

Thanks for the commentary. We have modified the text to clarify the significance.

Original manuscript:

CYP2D6 intermediate metabolizers showed higher elimination half-life (t1/2) and incidence of adverse reactions (ADRs) than normal plus ultrarapid metabolizers (univariate p-value (puv)=0.018; puv=0.008, multivariate p (pmv)=0.010, respectively).

Revised manuscript:

CYP2D6 intermediate metabolizers showed higher elimination half-life (t1/2) (univariate p-value (puv)=0.018) and incidence of adverse reactions (ADRs) (puv=0.008, multivariate p (pmv)=0.010) than normal plus ultrarapid metabolizers (univariate p-value (puv)=0.018; puv=0.008, multivariate p (pmv)=0.010, respectively).

  1. Page 3, lines 1-2 from the top: Test formulation (T) and reference formulation ( R) are mentioned. Indicate, which exactly trial/condition/dose is T and which is R in the Table 1.

Thank you for the commentary. In order to analyze their bioequivalence, both formulations ought to be administered under the same conditions. Therefore, in each clinical trial, both formulations were administered to every volunteer. The table legend in Table 1 has been modified as follows to clarify this information.

Original manuscript:

*Sample size is shown as total number (number of males/ number of females). BE: bioe-quivalence, PG: pharmacogenetic.

Revised manuscript:

*Sample size is shown as total number (number of males/ number of females). BE: bioe-quivalence, PG: pharmacogenetic. In each clinical trial, both formulations (T and R) were administered to every volunteer under the conditions and at the dose specified.

  1. Page 4, line 1 in the top. Define formally Tmax and C max in this chapter.

Thanks for the observation. We have modified the text as follows:

Original manuscript:

The tmax and Cmax (Cmax-ss in steady state for clinical trial C) were directly obtained from the plasma concentration-time curves.

Revised manuscript:

The maximum plasma concentration (Cmax; Cmax-ss in steady state for clinical trial C) and the time to reach it (tmax) were directly obtained from the plasma concentration-time curves.

  1. In the text, immediately before the Table 2 significances p<0.001 are indicated, then p = 0.005 and p=0.01, but in the Table 2 “*” indicates p<0.05. May be indicating lower p in the Table legend will be in favor of the authors (in case the information in the text is correct).

Thank you for you commentary. We have modify the significance in the table legend to p<0.01.

  1. Table 2, 3rdline: Height, the value in parentless 0.87 is typo or error, I think it must be 0.087

Thank you for your observation. Indeed, the value should be 0.087, rounded up to 0.09. The text has been amended accordingly.

  1. Table 3, 2rdline: CL/F, the value 20.03 is typo or error, I think it must be 2.003

Thank you for your observation. The text has been amended accordingly.

  1. Thus carefully double check all data in the Tables for the typos.

Thanks for the commentary. The data from the tables in the manuscript and in the supplementary material has been revised and some errors have been detected and corrected.

  1. Table 4 title: actual “Statistically significant associations between genotypes or phenotypes and …” may be just “Associations between genotypes or phenotypes and …”, because not all associations are significant in this table.

Thanks for the observation. The title of the table has been modified as follows:

Original manuscript:

Table 4. Statistically significant associations between genotypes or phenotypes and phar-macokinetic parameters.

Revised manuscript:

Table 4. Statistically significant Associations between genotypes or phenotypes and pharmacokinetic parameters.

  1. Page 4, line 6 from the bottom. “ADRs were present in ten volunteers…”. It is not clear immediately, these 10 cases referred for A, B or C trial? Indicate it in the text (even, then after it could be concluded from the Table 10).

Thanks you for the commentary. We have modified the text as follows:

Original manuscript:

ADRs were present in ten volunteers (12.66%), five of which suffered one ADR, three suffered two ADRs, and two suffered three ADRS.

Revised manuscript:

ADRs were present in ten volunteers (12.66%) in total (two in clinical trials A and C, and six in clinical trial B, Table 5), five of which suffered one ADR, three suffered two ADRs, and two suffered three ADRS.

  1. Discussion, page 10, line 18. Literature indicate that UGT1A4 have a negligible role in mitabegron metabolism [5], but in this paper authors show UGT1A4 to have a role in mitabegron pharmacokinetics. Please, comment/speculate more deeply this.

Thank you for your commentary. Although our results suggest that UGT1A4 might be involved in mirabegron metabolism, the lack of information available in literatura makes necessary further research to clarify the role of this enzyme on mirabegron pharmacokinetics. We have modified the text deepening in this possible association.

Original manuscript:

Nevertheless, other enzymes from the UGT1A gene cluster do participate in mirabegron metabolism, and it is known that enzymes from this family present substrate overlap [50]. However, a different study determined that the role of UGT1A4 in mirabegron metabolism is negligible [51]. Secondly, UGT1A3 rs2008584 was only associated with differences in tmax, which is a pharmacokinetic parameter more dependent on transporters activity, rather than that of hepatic metabolizing enzymes. In light of these considerations and the results obtained, it could be proposed that UGT1A4 rather than UGT1A3 might be involved in mirabegron metabolism, since genetic variation in UGT1A4 altered mirabegron t1/2. Nevertheless, further research on this association and on the impact of genetic variation in UGT1A4 should be performed.

Revised manuscript:

Nevertheless, other enzymes from the UGT1A gene cluster do participate in mirabegron metabolism, and it is known that enzymes from this family present substrate overlap [50]. Therefore, it could be proposed that the G allele in UGT1A4 rs2011425 causes a decreased mirabegron metabolization compared to the T allele, increasing the time needed for its elimination. However, a different study determined that the role of UGT1A4 in mirabegron metabolism is negligible [51]. Secondly, UGT1A3 rs2008584 was only associated with differences in tmax, which is a pharmacokinetic parameter more dependent on transporters activity, rather than that of hepatic metabolizing enzymes. In light of these considerations and the results obtained, it could be proposed that UGT1A4 rather than UGT1A3 might be involved in mirabegron metabolism, since genetic variation in UGT1A4 altered mirabegron t1/2. Nevertheless, further research on this association and on the impact of genetic variation in UGT1A4 activity should be performed.

  1. For the references [5], [21], please provide the date non only for the availability, but also the date of original reports update (if these reports from 2000, 2010, 2024?)

Thank you for the observation. The text has been modified accordingly.

Original manuscript:

[5] European Medicines Agency Betmiga. Accessed on: 12th April 2024. Available online: https://www.ema.europa.eu/en/medicines/human/EPAR/betmiga

 [21] World Medical Association World Medical Association Declaration of Helsinki Ethical Principles for Medical Research Involving Human Subjects. Accessed on 20th May 2024. Available online: https://jamanetwork.com/journals/jama/fullarticle/1760318

Revised manuscript:

[5] European Medicines Agency Betmiga. Last update: 15th October 2015. Accessed on: 12th April 2024. Available online: https://www.ema.europa.eu/en/medicines/human/EPAR/betmiga

[21] World Medical Association World Medical Association Declaration of Helsinki Ethical Principles for Medical Research Involving Human Subjects. Last update: 19th October 2013. Accessed on 20th May 2024. Available online: https://jamanetwork.com/journals/jama/fullarticle/1760318

  1. Reference [21] The phrase “World medical association” looks doubled?

Thank you for the observation. The text has been modified accordingly.

Reviewer 2 Report

Comments and Suggestions for Authors

I find the manuscript interesting and believe it will be of interest to a broad audiance as pharmacogenomic field is Rising, so it is important not only for general practitioners but also for scientists in the field of public health (pharmacovigilance), pharmacists and educators who present to their students up to date information on this matter. I find the study comprehensively conducted, methods extensively described which adds to the reproducibility and results well presented. The tables are presenting all the relevant data. The references seem appropriate and up to date. The conclusions of the manuscript are based on authors findings. I believe this study does add to the body of literature in the field of pharmacogenomics. I believe This manuscript could be used as literature for future CPIC guidelines.

Author Response

I find the manuscript interesting and believe it will be of interest to a broad audiance as pharmacogenomic field is Rising, so it is important not only for general practitioners but also for scientists in the field of public health (pharmacovigilance), pharmacists and educators who present to their students up to date information on this matter. I find the study comprehensively conducted, methods extensively described which adds to the reproducibility and results well presented. The tables are presenting all the relevant data. The references seem appropriate and up to date. The conclusions of the manuscript are based on authors findings. I believe this study does add to the body of literature in the field of pharmacogenomics. I believe This manuscript could be used as literature for future CPIC guidelines.

Thank you very much for your commentary. We appreciate the good valoration of our manuscript. We sincerely hope that our pharmacogenetic research will be useful in the clinical implementation of this discipline.
